# Improving Methanol Utilization by Reducing Alcohol Oxidase Activity and Adding Co-Substrate of Sodium Citrate in *Pichia pastoris*

**DOI:** 10.3390/jof9040422

**Published:** 2023-03-29

**Authors:** Shufan Liu, Haofan Dong, Kai Hong, Jiao Meng, Liangcai Lin, Xin Wu

**Affiliations:** 1Key Laboratory of Industrial Fermentation Microbiology, Ministry of Education, College of Bioengineering, Tianjin University of Science and Technology, Tianjin 300457, China; liushf@tib.cas.cn (S.L.); donghf@tib.cas.cn (H.D.); 2Laboratory of Nutrient Resources and Synthetic Biology, Tianjin Institute of Industrial Biotechnology, Chinese Academy of Sciences, Tianjin 300308, China; hongkai@tib.cas.cn (K.H.); wuxin@tib.cas.cn (X.W.)

**Keywords:** *Pichia pastoris*, alcohol oxidase activity, methanol utilization, sodium citrate, molecular mechanism

## Abstract

Methanol, which produced in large quantities from low-quality coal and the hydrogenation of CO_2_, is a potentially renewable one-carbon (C1) feedstock for biomanufacturing. The methylotrophic yeast *Pichia pastoris* is an ideal host for methanol biotransformation given its natural capacity as a methanol assimilation system. However, the utilization efficiency of methanol for biochemical production is limited by the toxicity of formaldehyde. Therefore, reducing the toxicity of formaldehyde to cells remains a challenge to the engineering design of a methanol metabolism. Based on genome-scale metabolic models (GSMM) calculations, we speculated that reducing alcohol oxidase (AOX) activity would re-construct the carbon metabolic flow and promote balance between the assimilation and dissimilation of formaldehyde metabolism processes, thereby increasing the biomass formation of *P. pastoris*. According to experimental verification, we proved that the accumulation of intracellular formaldehyde can be decreased by reducing AOX activity. The reduced formaldehyde formation upregulated methanol dissimilation and assimilation and the central carbon metabolism, which provided more energy for the cells to grow, ultimately leading to an increased conversion of methanol to biomass, as evidenced by phenotypic and transcriptome analysis. Significantly, the methanol conversion rate of AOX-attenuated strain PC110-AOX1-464 reached 0.364 g DCW/g, representing a 14% increase compared to the control strain PC110. In addition, we also proved that adding a co-substrate of sodium citrate could further improve the conversion of methanol to biomass in the AOX-attenuated strain. It was found that the methanol conversion rate of the PC110-AOX1-464 strain with the addition of 6 g/L sodium citrate reached 0.442 g DCW/g, representing 20% and 39% increases compared to AOX-attenuated strain PC110-AOX1-464 and control strain PC110 without sodium citrate addition, respectively. The study described here provides insight into the molecular mechanism of efficient methanol utilization by regulating AOX. Reducing AOX activity and adding sodium citrate as a co-substrate are potential engineering strategies to regulate the production of chemicals from methanol in *P. pastoris*.

## 1. Introduction

With the continuous expansion in the scale of chemical production, a series of problems, such as the shortage of fossil resources, the greenhouse effect, etc., have emerged one after another [1,2]. Growing environmental concerns and energy crises are driving us to think about how to move from a traditional petrochemical economy to sustainable feedstocks. The assimilation of one-carbon (C1) compounds by microorganisms has been proven as a promising approach to achieve sustainable development [3,4,5,6]. Among all C1 compounds, methanol has been identified to be an ideal and renewable feedstock for biomanufacturing [5,6,7]. Massive amounts of methanol can be produced not only from low-quality coal [8,9], but also from CO_2_ by photocatalytic or electrical reduction, which may contribute to the creation of process chains yielding value-added chemicals with (nearly) zero CO_2_ footprint [10,11,12]. Furthermore, as a liquid, methanol is easy to store and transport compared to gaseous C1 compounds, and its high degree of reduction is more beneficial for product biosynthesis than traditional sugar-based feedstocks [13]. Thus, the abundance, compatibility with current transportation and fermentation infrastructure, and high reduction potential result in the strong competitiveness of methanol as a substrate for biosynthesis [9].

The methylotrophic yeast *Pichia pastoris* (also known as *Komagataella phaffii*) is an ideal host for methanol-based biomanufacturing due to its natural methanol metabolism [14]. Methanol is first oxidized to hydrogen peroxide and formaldehyde by alcohol oxidase (AOX), both of which are compounds that are highly toxic to the cells. Hydrogen peroxide can be further decomposed into water and oxygen by catalase (CAT). Formaldehyde then enters either the dissimilation or assimilation branch of the methanol metabolism [15,16]. In the dissimilatory branch, formaldehyde is oxidized to CO_2_ under the action of S-(hydroxymethyl) glutathione dehydrogenase (Fld) and formate dehydrogenase (Fdh), yielding NADH [17,18,19]. In the assimilatory branch, formaldehyde reacts with xylulose-5-phosphate (Xu5P) under the catalysis of dihydroxyacetone synthase (Das) to generate glyceraldehyde-3-phosphate (GAP) and dihydroxyacetone (DHA), which then participates in the central carbon metabolism [16,20,21] (Figure 1). So far, *P. pastoris* has been widely used in the synthesis of various chemicals from methanol, such as lactic acid [22], non-animal chondroitin sulfate [23], malic acid [24], free fatty acids [25], etc. However, the methanol utilization efficiency remains low, which is not conducive to the synthesis of target products. Studies have shown that the flux of formaldehyde entering the dissimilation pathway of *P. pastoris* accounts for about 50–80% of the entire metabolic flow [26], resulting in carbon loss in the methanol metabolism. Further research indicates that formaldehyde’s toxicity is one of the primary obstacles limiting the carbon loss encountered by *P. pastoris* [25]. Thus, more efforts are still needed to solve the problem of the excessive accumulation of toxic formaldehyde in the process of methanol utilization. A recent report showed that the reduced AOX activity after long-term adaptation increased growth rates of *P. pastoris* [27]. This finding leads us to suspect that weakening AOX activity may reduce the toxicity of intracellular formaldehyde, leading to improved methanol utilization. Up to now, there is limited information connecting AOX activity, formaldehyde toxicity, and methanol utilization.

In this study, *P. pastoris* strains with reduced AOX activity were obtained by point mutation of alcohol oxidase 1 (AOX1) or the deletion of alcohol oxidase 2 (AOX2). GSMM calculations showed that reducing AOX activity could improve methanol utilization efficiency by redistributing the carbon metabolism in *P. pastoris*. By weakening the AOX activity, the accumulation of intracellular formaldehyde was reduced. Further transcriptome analysis showed that the reduced AOX activity upregulated the methanol dissimilation, assimilation and central carbon metabolism, which could provide more energy for cell growth. As a result, the reduced AOX activity led to an increase in the conversion of methanol to biomass. In addition, we also proved that adding a co-substrate of sodium citrate could further improve the central carbon metabolism and promote the formation of higher biomass in the AOX-attenuated strain in methanol-minimal medium. The endeavors described here provide insight into the mechanism of efficient methanol utilization by regulating AOX activity and adding sodium citrate, which are potential engineering strategies to regulate the production of chemicals from methanol in *P. pastoris*.

## 2. Materials and Methods

### 2.1. Strains, Media, and Cultivation

All plasmids and strains used in this study are listed in Table 1. *P. pastoris* PC110 and plasmid pPICZ-Cas9-gGUT1 [28] were provided by Prof. Zhou from Dalian Institute of Chemical Physics (DICP), Chinese Academy of Sciences. Other *P. Pastoris* strains and plasmids were constructed in this work. Unless otherwise specified, *P. pastoris* strains were cultivated in YPD medium consisting of 20 g/L glucose, 20 g/L peptone, and 10 g/L yeast extract at 30 °C at 220 rpm. In order to screen transformants, 100 μg/mL zeocin was supplemented into YPD medium. The Delft basic salt medium used for cell cultivation was consisted of 14.4 g/L KH_2_PO_4_, 7.5 g/L (NH_4_)_2_SO_4_, 0.5 g/L MgSO_4_·7H_2_O, 1 mL/L vitamin solution, 40 mg/L histidine, and 2 mL/L trace metal solution [29,30]. *Escherichia coli* DH5α used as the host for plasmid construction were cultivated at 37 °C at 220 rpm in Luria-Bertani (LB) consisting of 5 g/L yeast extract, 10 g/L tryptone, and 10 g/L NaCl, and 50 μg/mL zeocin was added to maintain plasmids.

### 2.2. Plasmids and Strains Construction

The method used for the genetic manipulation of *P. pastoris* was based on the utilization of CRISPR-Cas9-meditated genome editing system, as previously described [28]. To construct gRNA expression plasmid, 20 bp target sequences of gRNAs for genome targeting were designed using a user-friendly online web tool (CRISPR RGEN Tools, http://www.rgenome.net/) (accessed on 1 May 2022). For construction of gRNA plasmid for genome targeting of AOX1, the Cas part and gRNA part were obtained via PCR amplification with primers gAOX1-1F/gAOX1-1R and gAOX1-2F/gAOX1-2R using the plasmid pPICZ-Cas9-gGUT1(containing Cas9 and GUT1-gRNA under the control of bidirectional promoter P_HTX1_) as a template. The two parts were then fused by Gibson assembly to yield plasmid pPICZ-Cas9-gAOX1. For the construction of a long fragment with point mutation of AOX1, the upstream part and downstream part with an amino acid mutation at position 464 (R464 to K464) were amplified from the *P. pastoris* genome using primers AOX1-464-Up-F/AOX1-464-Up-R and AOX1-464-Down-F/AOX1-464-Down-R, respectively. The upstream and downstream fragments were then fused and amplified by fusion PCR with primers AOX1-464-Up-F/AOX1-464-Down-R, yielding long fragments with an amino acid mutation at position 464 of AOX1. The long fragment was further used as template to construct donor DNA after the first round of PCR using primers AOX1-Up-F/AOX1-Up-R and AOX1-Down-F/AOX1-Down-R and the second round of fusion PCR using primers AOX1-Up-F/AOX1-Down-R, respectively. The plasmid pPICZ-Cas9-gAOX1 and donor DNA with specific mutation were co-transformed into the competent cells of *P. pastoris*, and the transformed cells were grown for three days on YPD agar plates containing 100 μg/mL Zeocin. The mutants were verified by gene sequencing to determine the correct point mutation.

As for the deletion of AOX2, the plasmid pPICZ-Cas9-gAOX2 was constructed based on the methods described above using primers gAOX2-1F/gAOX2-1R and gAOX2-2F/gAOX2-2R. To construct donor DNA, fragments upstream and downstream of the *AOX2* gene were amplified from the *P. pastoris* genome using primers AOX2-Up-F/AOX2-Up-R and AOX2-Down-F/AOX2-Down-R, respectively. The upstream and downstream fragments were fused and amplified by fusion PCR with primers AOX2-Up-F/AOX2-Down-R. The pPICZ-Cas9-gAOX2 and donor DNA were then co-transformed into the competent cells of *P. pastoris*. Transformations were screened on YPD agar plates with 100 μg/L Zeocin. The AOX2 mutants were verified by PCR and further confirmed by gene sequencing. All primers used for strain and plasmid construction are listed in Table 2.

### 2.3. Growth Tests in 48-Well Plates

*P. pastoris* strains grown overnight in Delft + 2% (*w*/*v*) glycerol were centrifuged at 5000× *g* for 5 min at 4 °C. The supernatant was removed and the cell pellet was resuspended in Delft minimal medium. The cells were then inoculated into a 48-well plate containing 1 mL Delft + 0.5% (*v*/*v*) methanol medium with an initial OD_600_ of 0.25 at 30 °C at 800 rpm. Delft + 0.5% (*v*/*v*) methanol medium was used as blank control. Cell growth was evaluated by an automatic microbial growth curve analyzer and OD_600_ was measured per hour. Six replicates per condition were used.

### 2.4. Formaldehyde Quantification

Cells grown to the mid-log phase were collected by centrifugation at 5000× *g* for 10 min at 4 °C, washed three times and resuspended in 1 mL PBS buffer (pH = 7.4). An equal volume of fungal lysis buffer (0.5% *v*/*v* Triton x100, 1% *w*/*v* SDS, 100 mM NaCl, 1 mM EDTA, 10 mM Tris, pH = 8) was added and then cells were incubated at 37 °C for 30 min. The supernatants were obtained by centrifugation at 13,000 rpm for 5 min and analyzed for formaldehyde. The intracellular formaldehyde concentration was measured based on the previously described method [31]. Briefly, 50 μL of supernatant and 150 μL of NASH reagent (5 M ammonium acetate, 50 mM acetyl acetone, 135 mM acetic acid) were added into a 96-well plate. The plate was incubated at 37 °C for 10 min and the absorbance was read at 414 nm. The formaldehyde concentrations were expressed as μM/OD.

### 2.5. Alcohol Oxidase Activity Determination

As mentioned above, after cells were lysed, the supernatants were obtained by centrifugation at 3000 rpm for 25 min at 4 °C and used directly for AOX activity assays. AOX activity assays were performed using the Acetaldehyde Oxidase ELISA Kit (Meimian, Yancheng, China). Briefly, 5 μL of the previously prepared supernatant was added to the microtiter plate, followed by 45 μL of sample diluent. Next, 100 μL of HRP solution (2 mg/mL in HQ-H_2_O) was added into each well, and the plate was sealed with a sealing film, and incubated at 37 °C for 60 min. The liquid was discarded, and the solution was washed five times with 300 μL of wash solution and pat dry. Fifty microliters of each of the two color-developing reagents was added. The mixture was gently shaken and mixed and left at 37 °C for 15 min in the dark for color development. Following this, 50 μL of 2 M sulfuric acid (containing 0.1 M sodium sulfite) was added to stop the reaction, at which point the blue turned yellow, and the absorbance was measured at 450 nm [26]. AOX activity was calculated based on the prepared standard curve.

### 2.6. The Biomass and Methanol Conversion Rate Determination

The *P. pastoris* cells were grown in 250 mL shake flasks with 50 mL of Delft basic salt medium containing 0.5% methanol at 30 °C in a rotatory shaker at 220 rpm. To avoid methanol evaporation, the shake flask was covered with a sealing membrane. The biomass was characterized by dry cell weight (DCW) during the platform period (at this point the methanol had been completely consumed). The 50 mL sample was collected and then centrifuged at 5000× *g* at 4 °C for 10 min. The samples were washed three times with sterile water. The tubes containing centrifuged cells were dried at 105 °C until a constant weight was maintained. The methanol conversion rate was expressed as g DCW/g.

### 2.7. Intracellular NADH and NAD^+^ Quantitation

The intracellular NADH and NAD^+^ were extracted and determined using Solarbio^®^ Coenzyme I NAD(H) Content Assay Kit (Beijing Solarbio Science & Technology Co., Ltd., Beijing, China). Cells were collected by centrifugation at 5000× *g* for 10 min at 25 °C and washed three times with 1 mL PBS buffer (pH = 7.4). After that, cells were resuspended in 500 μL acid extraction buffer for NAD^+^ or 500 μL base extraction buffer for NADH and then disrupted by ultrasonication at 200 W for 1 min. The lysed cells were boiled for 5 min, and the supernatants were obtained by centrifugation at 10,000× *g* for 10 min at 4 °C. A 200 μL sample was taken of the supernatant to be mixed with the same volume of base extraction buffer for NAD^+^ or 200 μL of the supernatant was taken and mixed with the same volume of acid extraction buffer for NADH; then, the samples were centrifuged again under the same conditions and the supernatant was kept on ice for detection. Finally, after NADH reduced oxidized thiazolyl blue (MTT) to formazan through the hydrogen transfer effect of PMS, the absorbance was detected at 570 nm [25,32]. NAD^+^ can be reduced to NADH by alcohol dehydrogenase and further detected using the MTT reduction method.

### 2.8. Transcriptome Analysis

The control strain PC110 and the mutated strain PC110-AOX1-464 were grown to exponential phase in Delft + 0.5% (*v*/*v*) methanol basic salt medium at 30 °C. The total RNA was extracted from the cells by using TRIzol (Invitrogen, Carlsbad, CA, USA) according to the manufacturer’s protocols. RNA with integrity of more than 6.5 as detected by Agilent 2100 Nano (Agilent Technologies, Santa Clara, CA, USA) was adopted to perform library construction and sequencing. The Illumina Hi-Seq 2000 platform was used to construct and sequence the complementary DNA libraries at the Meiji Biotechnology (Shanghai) Co., Ltd. RESM software (https://deweylab.github.io/RSEM/) (accessed on 10 August 2022) was used to quantify the gene expression levels, yielding the transcripts per million reads (TPM). DEGseq (https://bioinfo.au.tsinghua.edu.cn/software/degseq) (accessed on 10 August 2022) was used to analyze the differential expression among the three samples. Genes with |log2FC| ≥ 1 and *p* < 0.05 were identified as significantly differentially expressed genes (DEGs). The KOBAS (https://kobas.cbi.pku.edu.cn/home.do) (accessed on 10 August 2022) was used to perform the Kyoto Encyclopedia of Genes and Genomes (KEGG) enrichment analysis.

### 2.9. Statistical Analysis

One-way analysis of variance was implemented in SPSS for Windows 10.0 (SPSS, Inc., Chicago, IL, USA).

## 3. Results

### 3.1. Construction of P. pastoris Strains with Reduced AOX Activity by Point Mutation of AOX1 or Deletion of AOX2

In order to solve the problem of intracellular formaldehyde toxicity in the process of methanol utilization, we focused on the regulation of AOX, the first enzyme in the methanol metabolism pathway, which catalyzes the conversion of methanol into formaldehyde. *P. pastoris* harbors two genes for the alcohol oxidase gene, *AOX1* and *AOX2*. The vast majority of the AOX enzymes are expressed by AOX1, despite the regulation pattern of the AOX2 gene being exactly the same as the AOX1 gene. Previous studies have shown that impairing AOX1 function led to slower growth, whereas the reduced AOX1 activity resulted in the increased growth rates on methanol [27,33,34]. Accordingly, we hypothesized that AOX1, relative to AOX2, is essential for methanol utilization in *P. pastoris* and that the AOX activity can be attenuated by weakening AOX1 or knocking out AOX2, thereby reducing intracellular formaldehyde accumulation. In this study, a point mutation of AOX1 (R464 to K464) (Figure 2a) and deletion of AOX2 were implemented in the wild-type strain PC110, creating strains PC110-AOX1-464 and PC110-ΔAOX2, respectively. As expected, both mutant strains exhibited lower AOX activity compared with the control strain PC110 (Figure 2b). The AOX activity of the PC110-AOX1-464 (up to % AOX activity of 96.8) and PC110-ΔAOX2 (up to % AOX activity of 137) decreased by 34% and 7%, respectively, compared to the corresponding measurement for PC110 (up to % AOX activity of 147). All these data suggest that the *P. pastoris* strains with reduced AOX enzyme activity can be obtained from mutant AOX1 or the knockout of AOX2 and that mutant AOX1 has a greater effect on reducing AOX activity.

### 3.2. Data of CO_2_ Loss and Biomass Accumulation of P. pastoris with Reduced AOX Activity via Genome-Scale Metabolic Model (GSMM) Calculations

GSMM can predict the phenotype of a microorganism in a range of conditions, including those derived from genetic modification [35]. In this study, a new version of GSMM (v3.0) was used to predict the biomass of *P. pastoris*, as well as the rate of CO_2_ production with reduced AOX activity in the methanol-minimum medium. Firstly, the methanol utilization rates of PC110, PC110-AOX1-464, and PC110-ΔAOX2 were set as 10, 6.6, and 9.3 mmol/gDcw/h, respectively, according to the measured activity of AOX above, and then Flux Balance Analysis (FBA) was performed (Table 3). Metabolic flux distribution was always sensitive to biomass composition [36]. The flux distribution calculation showed that the CO_2_ generation rate of PC110 is 6.83 mmol CO_2_/mmol methanol, the CO_2_ generation rate of PC110-AOX1-464 is 4.98 mmol CO_2_/mmol methanol, and the CO_2_ generation rate of PC110-ΔAOX2 is 6.45 mmol CO_2_/mmol methanol; the corresponding biomass generation percentages are 2.56%, 2.62%, and 2.57%, respectively. The above GSMM calculation results indicate that the conversion of methanol to biomass is increased due to the decrease in CO_2_ release after reducing AOX activity, especially mutant AOX1. The calculation results of GSMM showed that the internal carbon metabolic flow of *P. pastoris* was derived to conduct a thorough reconstructing, ultimately leading to increased biomass accumulation after the reduction in AOX activity.

### 3.3. Reduced AOX Activity Decreased Intracellular Formaldehyde Accumulation and Increased Cell Growth on Methanol in P. pastoris

As the first intermediate of methanol utilization, formaldehyde confers the main toxicity and impairs cellular robustness [37]. It has been reported that the metabolic accumulation of formaldehyde increased the mortality of the cells [13]. There were engineering efforts to overcome this challenge, including the peroxisomal compartmentalization strategy for engineering the methanol utilization pathway of *P. pastoris* and enhancing methanol metabolism by balancing the assimilation pathway [38,39]. However, little research has been conducted to reduce the toxicity of formaldehyde by weakening the activity of AOX. In this study, both mutant strains showed lower intracellular formaldehyde accumulation compared to the control strain PC110. The PC110-AOX1-464 and PC110-ΔAOX2 strains accumulated low levels of formaldehyde at 376 μM/OD and 504 μM/OD, respectively, compared to a relatively high level of formaldehyde in PC110 cells up to 547 μM/OD (Figure 3a). Furthermore, it was found that both mutants acquired higher cell density in the methanol-minimal medium, which was consistent with the expected result of the GSMM calculation. The final biomass of strains PC110-AOX1-464 (up to OD_600_ of 3.76) and PC110-ΔAOX2 (up to OD_600_ of 3.35) increased by 19% and 6%, respectively, relative to the control strain PC110 (up to OD_600_ of 3.17) in 0.5% methanol-minimal medium (Figure 3b). All these findings suggest that reduced AOX activity, especially that of mutant AOX1, could decrease the intracellular accumulation of formaldehyde and thereby promote cell growth in methanol-minimal medium.

### 3.4. Reduced AOX Activity Promoted the Methanol Utilization Efficiency in P. pastoris

As we mentioned in the Introduction section, 50–80% of the methanol is converted into CO_2_ through the dissimilation pathway, leading to low methanol utilization [26]. Based on the current knowledge, formaldehyde’s toxicity is supposed to be one of the primary obstacles limiting carbon loss encountered by *P. pastoris* [25]. We hypothesized that reducing the accumulation of intracellular formaldehyde by weakening AOX could promote methanol utilization efficiency, and this seems to have been confirmed by the results of the GSMM calculations and growth data from 48-well plates (Figure 3b). However, the 48-well plate data cannot really reflect the actual utilization efficiency of methanol because of the methanol volatilization during the oscillation. To avoid methanol evaporation, the conversion of methanol to biomass was performed in the 250 mL shake flasks (the filling volume was 50 mL) covered with a sealing membrane in 0.5% methanol minimal medium. As shown in Table 4, the final biomass (DCW) of the PC110-AOX1-464 strain reached 0.144 g with a methanol conversion rate of 0.364 g DCW/g, representing a 14% increase compared to the control strain PC110 (*p* < 0.01). However, there was no significant change between the PC110-ΔAOX2 strain and the PC110 strain in terms of cell dry weight and methanol conversion rate (*p* > 0.05). By combining these results with the 48-well plate data, we concluded that the reduction in AOX activity following the point mutation of AOX1 decreased the intracellular accumulation of formaldehyde, which ultimately promoted methanol utilization efficiency in *P. pastoris*.

### 3.5. Transcriptomics Analysis Revealed the Mechanism of Enhanced Methanol Utilization Associated with Reduced AOX Activity in P. pastoris

Since PC110-AOX1-464 strain was superior to the PC110-ΔAOX2 strain in methanol utilization rate, a whole-transcriptome analysis was conducted to reveal the mechanism of enhanced methanol utilization associated with reduced AOX activity in the PC110-AOX1-464 strain relative to the control strain PC110. The RNA-Seq data indicated that for the mutant strain PC110-AOX1-464, 1197 genes were significantly differentially expressed, where 699 were upregulated and 498 were downregulated (Appendix A). Results based on KEGG enrichment analysis indicated that nine pathways were significantly enriched (*p* < 0.05), including methanol metabolism, glycolysis (EMP), citrate cycle (TCA cycle), and glutamate metabolism, etc. (Appendix A). As we expected, genes involved in formaldehyde assimilation, including *PAS_chr3_0834* (encoding DAS1, dihydroxyacetone synthase 1), *PAS_chr3_0832* (encoding DAS2, dihydroxyacetone synthase 2), *PAS_chr1-1_0072* (encoding Fba, fructose 1,6-bisphosphate aldolase), *PAS_chr3_0868* (encoding Fbp, fructose 1,6-bisphosphate), *PAS_chr3_0951* (encoding Tpi, triose phosphate isomerase), and *PAS_chr4_0212* (encoding Rpi, ribose-5-phosphate ketol-isomerase), were all upregulated. At the same time, genes involved in the pentose phosphate pathway (PPP), including *PAS_chr3_0277* (encoding Gnd, 6-phosphogluconate dehydrogenase), *PAS_chr1-4_0150* (encoding Tkta, Tktb, transketolase), *PAS_chr2-2_0337* (encoding TalA, TalB, transaldolase), and *PAS_chr2-2_0338* (encoding TalA, TalB, transaldolase), were also upregulated, possibly in order to provide more precursor Xu5P to drive the assimilation of formaldehyde (Figure 4a).

Unexpectedly, genes involved in formaldehyde dissimilation, including *PAS_chr3_1028* (encoding Fld1, S-(hydroxymethyl) glutathione dehydrogenase), *PAS_chr3_0867* (encoding Fgh1, S-formylglutathione hydrolase), and *PAS_chr3_0932* (encoding Fdh1, NAD^+^-dependent formate dehydrogenase), were also upregulated for the mutant strain PC110-AOX1-464 (Figure 4a). It is known that formaldehyde is toxic to cells. With the decrease in the accumulation of formaldehyde due to the reduced AOX activity, formaldehyde’s toxic effect on cells will also reduce. At this time, the environment is more conducive to cell growth. Furthermore, as described in previous research [40,41], the primary function of the dissimilation pathway may be energy provision rather than the detoxification of formaldehyde. Therefore, the upregulation of formaldehyde dissimilation in the PC110-AOX1-464 strain may be to provide more energy for cell growth in a methanol-minimal medium.

Furthermore, most of the genes involved in EMP, including *PAS_chr3_0951* (encoding Tpi, triose phosphate isomerase), *PAS_chr2-1_0437* (encoding GAPDH, glyceraldehyde-3-phosphate dehydrogenase), *PAS_chr1-4_0292* (encoding Pgk, 3-phosphoglycerate kinase), *PAS_chr3_0826* (encoding gpmA, tetrameric phosphoglycerate mutase), *PAS_chr3_0082* (encoding ENO, enolase I), *PAS_chr2-2_0294* (encoding AceE, E1 alpha subunit of the pyruvate dehydrogenase (PDH) complex), *PAS_chr1-4_0593* (encoding AceE, E1 beta subunit of the pyruvate dehydrogenase (PDH) complex), and *PAS_chr1-1_0050* (encoding Dlat, dihydrolipoamide acetyltransferase component (E2) of pyruvate dehydrogenase complex), were greatly upregulated; however, only the gene *PAS_chr1-1_0427* (encoding gpmB, putative protein of unknown function with some similarity to GPM1/YKL152C) was downregulated for the mutant strain PC110-AOX1-464. In addition, genes in TCA, including *PAS_chr1-1_0475* (encoding CS, hypothetical protein), *PAS_chr1-3_0104* (encoding ACO, aconitase), *PAS_chr1-1_0233* (encoding Idh1, Idh2, mitochondrial NADP^+^-specific isocitrate dehydrogenase), *PAS_chr2-1_0120* (encoding Idh3, subunit of mitochondrial NAD^+^-dependent isocitrate dehydrogenase), *PAS_chr4_0580* (encoding Idh3, subunit of mitochondrial NAD^+^-dependent isocitrate dehydrogenase), *PAS_chr3_0831* (encoding LSC1, alpha subunit of succinyl-CoA ligase), *PAS_chr4_0733* (encoding SdhA, flavoprotein subunit of succinate dehydrogenase), *PAS_chr1-4_0487* (encoding SdhC, putative protein of unknown function), *PAS_chr3_0647* (encoding FumC, fumarase), *PAS_chr2-1_0238* (encoding Mdh2, mitochondrial malate dehydrogenase), and *PAS_chr4_0815* (encoding Mdh2, mitochondrial malate dehydrogenase), were also greatly upregulated (Figure 4a). As mentioned above, increased cell growth requires more energy, so the upregulation of EMP and TCA pathways is in line with our expectations. Here, we observed the increased NAD^+^/NADH ration in strain PC110-AOX1-464, representing 1.96 times that of the control strain PC110 (Figure 4b). It was worth noting that genes involved in glutamate metabolism, including *PAS_chr1-1_0107* (encoding Gdn1, NADP^+^-dependent glutamate dehydrogenase) and *PAS_chr4_0785* (encoding Gln1, Glutamine synthetase), were also upregulated in the mutant strain PC110-AOX1-464. In yeast, nitrogenous substances that make up amino acids in biomass are mainly converted from glutamate and glutamine [42]. Glutamate and glutamine, together with α-ketoglutarate, link the TCA cycle and nitrogen metabolism. Thus, the upregulated glutamate metabolic pathway and TCA cycle will inevitably lead to the increase in biomass formation [43,44,45,46].

Collectively, the transcriptome data revealed that the reduced formaldehyde formation caused by weakening AOX activity upregulated the methanol dissimilation, assimilation and central carbon metabolism (including PPP, EMP, TCA cycle, glutamate metabolism). The reduced AOX activity could re-construct the balance of carbon metabolism, leading to greater energy generation, which ultimately resulted in an increased promotion of the conversion of methanol to biomass (Figure 4c).

### 3.6. Addition of Co-Substrate of Sodium Citrate Further Promoted Cell Growth and Biomass Accumulation of PC110-AOX1-464 in Methanol-Minimal Medium

According to the transcriptome data, energy deficiency and insufficient carbon metabolic flow greatly limited cell growth and biomass accumulation of *P. pastoris* in methanol-minimal medium. It was reported that sodium citrate can stimulate isocitrate dehydrogenase activity and strengthen the TCA cycle [47]. Adding sodium citrate could help to increase the intracellular energy content and improve the central carbon metabolic flow [48]. In this study, adding sodium citrate as a co-substrate was performed to further improve the conversion of methanol to biomass in the PC110-AOX1-464 strain. First, the effects of adding different concentrations of sodium citrate on methanol utilization was explored. Here, the dosage of sodium citrate was set as 0, 2, 4, 6, 8, and 10 g/L. It was found that the optimal concentration of sodium citrate was 6 g/L based on the cell growth of strain PC110-AOX1-464 under different concentrations of sodium citrate (Figure 5). The final biomass of PC110-AOX1-464 with 6 g/L sodium citrate (up to OD_600_ of 4.41) was 17% and 39% higher than PC110-AOX1-464 and PC110, respectively, without sodium citrate addition in 0.5% methanol-minimal medium. As shown in Table 5, the final biomass (DCW) of the PC110-AOX1-464 strain with 6 g/L sodium citrate reached 0.175 g with a methanol conversion rate of 0.442 g DCW/g, representing a 20% and 39% increase compared to PC110-AOX1-464 and PC110 without sodium citrate addition (*p* < 0.01), respectively. All these findings suggest that adding sodium citrate can further improve the central carbon metabolism and promote the biomass accumulation of strain PC110-AOX1-464 in methanol-minimal medium.

## 4. Discussion

Methanol, which can be used as the single carbon source for cell growth in *P. pastoris*, is low cost and has a high energy content [14,49,50]. Nowadays, considerable breakthroughs have been made in the synthesis of methanol via the hydrogenation of CO_2_ [12,51], opening an avenue for methanol to become a renewable raw material in the future of biomanufacturing [52]. However, its application in biobased manufacturing is constrained because of the accumulation of formaldehyde and formate in the methanol metabolism, which are both toxic intermediates [25]. In addition, formaldehyde may be more toxic to cells than formate, and this is one of the primary obstacles limiting carbon loss encountered by *P. pastoris* [26,37]. Therefore, the key to promote the development of methanol in biomanufacturing is to reduce the toxicity of formaldehyde and enhance the methanol utilization efficiency in *P. pastoris* [25].

In *P. pastoris,* peroxisome is the main site of the methanol metabolism. AOX monomers, expressed intracellularly, need to be led into the peroxisome and then assembled into the active enzyme [53]. It has been reported that AOX is the key enzyme in the formation of formaldehyde in the methanol metabolism pathway of *P. pastoris*. *P. pastoris* contains two alcohol oxidases, AOX1 and AOX2. AOX1 is the main alcohol oxidase in *P. pastoris*, and its content makes up more than 90% of the total AOX protein content, while AOX2 only plays a small role [33,34]. While AOX activity is certainly a prerequisite for the growth of *P. pastoris* in methanol, reducing AOX activity does not necessarily lead to reduced growth rates or fitness, which is supported by a recent finding that a lower AOX activity increased the cell growth rates of *P. pastoris* [27]. Based on the above analysis, the mutation of AOX1 and the deletion of AOX2 were carried out in this study to explore the effect of reduced AOX activity on methanol utilization in *P. pastoris* and its molecular mechanism.

In this study, GSMM calculations predicted that reducing AOX activity, especially via the point mutation of AOX1, could decrease CO_2_ production and increase biomass accumulation in methanol minimal medium. After experimental verification, we found that reducing AOX activity could decrease intracellular formaldehyde accumulation and could indeed promote the conversion of methanol to biomass. Further transcriptome analysis revealed that the decrease in AOX activity was accompanied by the upregulated dissimilation and assimilation of methanol and the central carbon metabolism, thereby providing more energy for cell growth in methanol. By combining the results of this study with previous findings [27], we concluded that the reduced AOX activity led to a decrease in intracellular formaldehyde, reduced formaldehyde-induced toxicity, and promoted balance between the assimilation and dissimilation of formaldehyde in metabolism processes. As far as we know, this is the first systematic study connecting AOX activity, formaldehyde toxicity, and methanol utilization rate. In addition, we also proved that adding a co-substrate of sodium citrate could further increase the central carbon metabolic flow and promoted the higher formation of biomass in the AOX-attenuated strain in a methanol-minimal medium. Knowledge gained from this study is that reduced AOX activity and the addition of sodium citrate can be used to construct a new balance in the carbon metabolic flow so that cell growth and biomass accumulation can be enhanced using methanol as a single carbon source in *P. pastoris*.

## 5. Conclusions

This study described here showed that reduced AOX activity could decrease formaldehyde toxicity and promote biomass accumulation of *P. pastoris* in methanol-minimal medium. In addition, we also proved that adding sodium citrate could further improve the central carbon metabolism and promoted the conversion of methanol to biomass in the AOX-attenuated strain. All these results indicate that reducing AOX activity and adding sodium citrate are potential strategies to increase methanol utilization in methanol-based *P. pastoris* cell factories. The strategies described here may also be suitable for engineer other native methylotrophic yeasts in methanol-based biomanufacturing.

## Figures and Tables

**Figure 1 jof-09-00422-f001:**
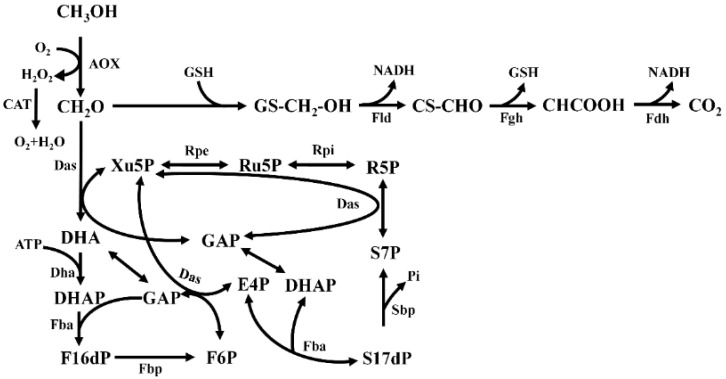
Sketch of methanol metabolism in *P. pastoris*. Enzymes: AOX, alcohol oxidase; Fld, formaldehyde dehydrogenase; Fgh, S-formylglutathione hydrolase; Fdh, formate dehydrogenase; Das, dihydroxyacetone synthase; DHA, dihydroxyacetone kinase; Fba, fructosebisphosphate aldolase/sedoheptulose-bisphosphate aldolase; Fbp, fructose bisphosphatase; Sbp, sedoheptulose bisphosphatase; Rpe, ribulose-phosphate 3-epimerase; Rpi, ribose-5-phosphate isomerase. Metabolites: Xu5P, xylulose-5-phosphate; Ru5P, ribulose-5-phosphate; R5P, ribose-5-phosphate; DHA, dihydroxyacetone; DHAP, dihydroxyacetone phosphate; F16dP, fructose-1,6-bisphosphate; F6P, fructose-6-phosphate; GAP, glyceraldehyde-3-phosphate; E4P, erythrose-4-phosphate; S7P, sedoheptulose-7-phosphate; S17dP, sedoheptulose-1,7-bisphosphate.

**Figure 2 jof-09-00422-f002:**
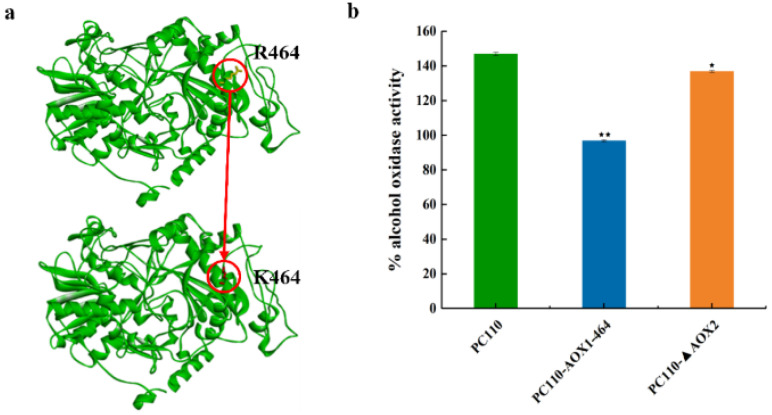
The structural change in AOX1 and the AOX activities of different *P. pastoris* strains. (**a**) The change in AOX1 structure between PC110 and PC110-AOX1-464. The mutated amino acid position was highlighted in red circle; (**b**) The AOX activities of PC110 strain and mutant strains, PC110-AOX1-464 and PC110-ΔAOX2, in 0.5% methanol minimal medium. Average values and standard deviations are the data of triplicate experiments. An asterisk indicates a significant difference with ^⋆^ *p* < 0.05, ^⋆⋆^ *p* < 0.01.

**Figure 3 jof-09-00422-f003:**
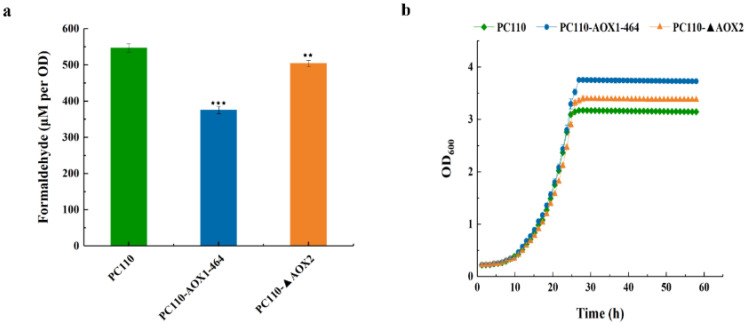
Determination of intracellular formaldehyde accumulation and cell growth of different *P. pastoris* strains in 0.5% methanol-minimal medium. (**a**) The intracellular accumulation of formaldehyde in PC110 strain and mutant strains, PC110-AOX1-464 and PC110-ΔAOX2; (**b**) The growth curves of PC110 strain and mutant strains, PC110-AOX1-464 and PC110-ΔAOX2. Average values and standard deviations are the data of triplicate experiments. An asterisk indicates a significant difference with ^⋆⋆^ *p* < 0.01, ^⋆⋆⋆^ *p* < 0.001.

**Figure 4 jof-09-00422-f004:**
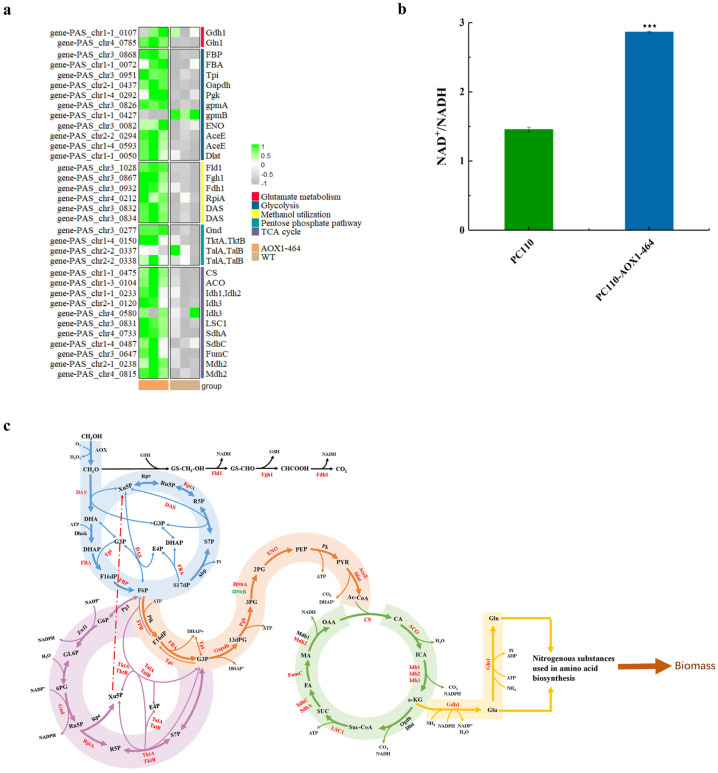
Change in intracellular NAD^+^/NADH ratio and DEGs between PC110 strain and PC110−AOX1−464 mutant strain. (**a**) Cluster analysis related to DEGs from methanol metabolism, PPP, EMP, TCA cycle, and glutamate metabolism; (**b**) The ratio of intracellular NAD^+^/NADH in PC110 strain and PC110−AOX1−464 mutant strain. Average values and standard deviations are the data of triplicate experiments. An asterisk indicates a significant difference, where ^⋆⋆⋆^ *p* < 0.001; (**c**) Sketch of DEGs in methanol metabolism, PPP, EMP, TCA cycle, and glutamate metabolism in PC110−AOX1−464 strain. Genes in black indicated no significant difference, genes in red were significantly upregulated, and genes in green were significantly downregulated.

**Figure 5 jof-09-00422-f005:**
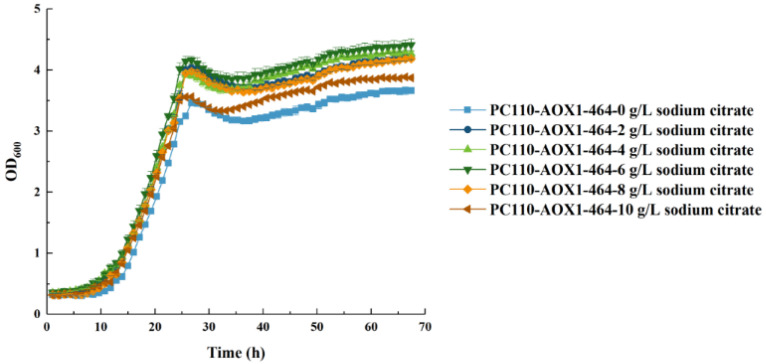
Cell growth of PC110-AOX1-464 strain in 0.5% methanol minimal medium with addition of different concentrations of sodium citrate. The dosage of sodium citrate was set as 0, 2, 4, 6, 8, and 10 g/L. Average values and standard deviations are the data of triplicate experiments.

**Table 1 jof-09-00422-t001:** Strains and plasmids used in this study.

Strains and Plasmids	Relevant Characteristics	Sources
** *P. pastoris* **		
PC110	Derived from GS115, P_GAP_-PpRAD52-T_AOX1_	[28]
PC110-AOX1-464	Derived from PC110, AOX1∷R464 to K464	This study
PC110-ΔAOX2	Derived from PC110, ΔAOX2	This study
** *E. coli* **		
DH5a	F^−^, φ80*lacZ* Δ*M15,* Δ*(lacZYA-argF)U169, deoR, recA1, endA1, hsdR17(rk^−^, mk^+^), phoA, supE44, λ^−^, thi-1, gyrA96, relA1*	Lab stock
**Plasmids**		
pPICZ-Cas9-gGUT1	ori, Amp, Zeocin, T_DAS1_-Cas9-P_HTX1_-GUT1-gRNA2-T_AOX1_	[28]
pPICZ-Cas9-gAOX1	ori, Amp, Zeocin, T_DAS1_-Cas9-P_HTX1_-AOX1-gRNA-T_AOX1_	This study
pPICZ-Cas9-gAOX2	ori, Amp, Zeocin, T_DAS1_-Cas9-P_HTX1_-AOX2-gRNA-T_AOX1_	This study

**Table 2 jof-09-00422-t002:** Primers used for strain and plasmid construction in this study.

Primer	Sequences (5′→3′)
gAOX1-1F	AACAGACTGATGAGTCCGTGAGGACGAAACGAGTAAGCTCGTCTCTGTTCCCATACTCATCCGGTTTTAGAGCTAGAAATAGCA
gAOX1-1R	ACGGGAAGTCTTTACAGTTT
gAOX1-2F	CTCCTAACTAAAACTGTAAAGACTTCCCGTTTAAACTTTTCTTTTCTTCT
gAOX1-2R	GTTTCGTCCTCACGGACTCATCAGTCTGTTTTTGATTTGTTTAGGTAACT
AOX1-464-Up-F	ACTTGCCAGGTGTCGGAAGA
AOX1-464-Up-R	TTTAGACTTCTTGTAAGCCC
AOX1-464-Down-F	CCTATGGTTTGGGCTTACAAGAAGTCTAAAGAAACCGCTAGAAGAATGGA
AOX1-464-Down-R	TTAGAATCTAGCAAGACCGGTC
AOX1-Up-F	GAGTCTTTCGATGACTTCGT
AOX1-Up-R	TTCGGATGAGTATGGGAACAGA
AOX1-Down-F	CACCACCCTCTGTTCCCATACTCATCCGAAGCCAGAGCCTTGGAAATGGA
AOX1-Down-R	CTTGAACTGAGGAACAGTCA
gAOX2-1F	CAGAGGCTGATGAGTCCGTGAGGACGAAACGAGTAAGCTCGTCCCTCTGGTTCTGCAAAGATCGTTTTAGAGCTAGAAATAGCA
gAOX2-1R	ACGGGAAGTCTTTACAGTTT
gAOX2-2F	CTCCTAACTAAAACTGTAAAGACTTCCCGTTTAAACTTTTCTTTTCTTCT
gAOX2-2R	GTTTCGTCCTCACGGACTCATCAGCCTCTGTTTGATTTGTTTAGGTAACT
AOX2-Up-F	TTAAGCGAAAGAGACAAGACAACGA
AOX2-Up-R	TTTTCTCAGTTGATTTGTTTGTGGG
AOX2-Down-F	AAATCCCCACAAACAAATCAACTGAGAAAATTTATGTTGTATCTATGAATATTTT
AOX2-Down-R	TTAGACTACTCTGAATCCGAGAAGA

The underlined are the modified bases.

**Table 3 jof-09-00422-t003:** Data of CO_2_ production rate and biomass accumulation of *P. pastoris* with reduced AOX activity calculated via GSMM.

	Parameter	PC110	PC110-AOX1-464	PC110-ΔAOX2
Input	Methanol utilization rate (mmol/gDcw/h)	10	6.6	9.3
Output	CO_2_ (mmol CO_2_/mmol methanol)	6.83	4.98	6.45
	Biomass (%)	2.56	2.62	2.57

**Table 4 jof-09-00422-t004:** The biomass and methanol utilization rate of PC110 strain and mutant strains, PC110-AOX1-464 and PC110-ΔAOX2, in 0.5% methanol-minimal medium.

Strains	DCW (g)	Methanol Utilization Rate (g DCW/g)
PC110	0.126 ± 0.002	0.318 ± 0.005
PC110-AOX1-464	0.144 ± 0.001	0.364 ± 0.003
PC110-ΔAOX2	0.128 ± 0.002	0.324 ± 0.006

**Table 5 jof-09-00422-t005:** The biomass and methanol utilization rate of PC110-AOX1-464 strain with addition of different concentrations of sodium citrate in 0.5% methanol-minimal medium.

PC110-AOX1-464 Strain with Addition of Different Concentrations of Sodium Citrate (g/L)	DCW (g)	Methanol Utilization Rate (g DCW/g)
0	0.146 ± 0.005	0.368 ± 0.007
2	0.167 ± 0.003	0.421 ± 0.007
4	0.170 ± 0.004	0.430 ± 0.007
6	0.175 ± 0.002	0.442 ± 0.006
8	0.167 ± 0.003	0.420 ± 0.008
10	0.155 ± 0.001	0.390 ± 0.003

## Data Availability

Data are available from the authors.

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
