# Peer review of "Improving Methanol Utilization by Reducing Alcohol Oxidase Activity and Adding Co-Substrate of Sodium Citrate in Pichia pastoris"

_jof, 2023, doi:10.3390/jof9040422_

Round 1

Reviewer 1 Report

General comments to manuscript number JOF 2250804:

The manuscript entitled: " Improving methanol utilization by reducing alcohol oxidase ac-2 tivity and adding co-substrate sodium citrate in Pichia pastoris " by  Shufan Liu et al, describe the production of AOX.

However the manuscript needs revision before being acceptable. The specific comments below should help the authors to improve the quality of the manuscript.

Specific comments to manuscript number JOF 22508041:

For all figures: the shade of green does not make it possible to clearly distinguish the different elements presented: put different colors.

Check que quality of the figure 3A and C.

Line 54-60: Give a figure with the various enzymatic reaction.

Table 2 : place the arrow between 5’ and 3’

Line 143 : P. Pastoris ?

Line 187-190 : Why use 2 types of buffer ? Acid or base? What is their use?

Line 213 : Windows 20.0 ?

Check que quality of the figure 3A and C.

Figure 4 and table 4 are very difficult to read : change que colors and simplify the fisrt column of the table.

Author Response

23 Mar, 2023

Dear Reviewer

Your comments on our manuscript “Improving methanol utilization by reducing alcohol oxidase activity and adding co-substrate sodium citrate in Pichia pastoris(Manuscript ID: jof-2250804) are highly appreciated. We have gone through the manuscript again and revised it carefully according to your suggestions. All the changes have been highlighted in red in the text. Thus the revised version is submitted online for you to reconsider for publication in Journal of Fungi. In the following pages are our point-by-point responses to each of the comments.

We look forward to hearing from you at your earliest convenience.

Yours sincerely,

Lin Liangcai

______________________________

Liangcai Lin (PhD, Associate professor)

Tianjin University of Science and Technology

College of Bioengineering, Tianjin 300457, China

E-mail: lclin@tust.edu.cn.

Responses to the comments of Reviewer #1:

  • 1-For all figures: the shade of green does not make it possible to clearly distinguish the different elements presented: put different colors;

Response: Thanks for your suggestion. We have redrawn all figures in different colors. Please see Figures 1-5 in the revised manuscript.

  • 2-Check que quality of the figure 3A and C;

Response: Thanks for your suggestion. We have checked and redrawn this figure, please see Figure 4A and C in the revised manuscript.

  • 3-Line 54-60: Give a figure with the various enzymatic reaction;

Response: Thanks for your suggestion. We have provided the figure with various enzymatic reaction of methanol metabolism and related description. Please see Figure 1 and lines 82-90 in the revised manuscript.

  • 4-Table 2: place the arrow between 5’ and 3’;

Response: It is revised. Please see Table 2 in the revised manuscript.

  • 5-Line 143: pastoris?

Response: Thanks for your correction. We have replaced “pastoris” with “P. pastoris”, please see line 159 in the revised manuscript.

  • 6-Line 187-190: Why use 2 types of buffer? Acid or base? What is their use? The second major issue is references;

Response: In this study, the intracellular NADH and NAD+ were extracted and determined by using So-larbio○R Coenzyme I NAD(H) Content Assay Kit (Beijing Solarbio Science& Technol-ogy Co., Ltd.). Two types of buffers were used because different buffers need to be used for different extracts. For NAD+, the acid extraction buffer should be added first and then added the base extraction buffer after the extraction of supernatant; while for NADH, the base extraction buffer should be added first and then added the acid extraction buffer after the extraction of supernatant. We described this method more clearly and updated references in the revised manuscript, please see lines 203-209 in the revised manuscript.

References:

Gao, J., Li, Y., Yu, W., and Zhou, Y.J. 2022. 'Rescuing yeast from cell death enables overproduction of fatty acids from sole methanol', Nature Metabolism, 4: 932-943.

Cai, P., Wu, X., Deng, J., Gao, L., Shen, Y., Yao, L., and Zhou, Y.J. 2022. 'Methanol biotransformation toward high-level pro-duction of fatty acid derivatives by engineering the industrial yeast Pichia pastoris', Proceedings of the National Academy of Sci-ences of the United States of America, 119: e2201711119.

  • 7-Line 213: Windows 20.0?

Response: Thanks for your correction. We have replaced “Windows 20.0” with “Windows 10.0”, please see line 230 in the revised manuscript.

  • 8- Figure 4 and table 4 are very difficult to read: change que colors and simplify the first column of the table;

Response: Thanks for your suggestions. It is revised, please see figure 5 and first column of the table 4 and table 5 in the revised manuscript.

Reviewer 2 Report

Dear authors,

I would like to congratulate you for the work presented here in this manuscript. The topic is of great interest and more then that,  you managed to present the experiments in a clear way, even though the study is very complex. 

Even though, the abstract must be significantly improve since it contains only very general statements and is not highlighting the main results that you revealed in the manuscript.

There are 2 things that in my opinion should be more clearly stated.

 In figure 1a, you presented a 3D structure, but you don't mention anything about it, is a simulation is from RCSB, change was made at the aminoacid level, did this change /point mutation affects the 3D structure?

2. You used a Delft medium for several cultivation experiments but I couldn't fine any reference or detailed composition.

Author Response

23 Mar, 2023

Dear Reviewer

Your comments on our manuscript “Improving methanol utilization by reducing alcohol oxidase activity and adding co-substrate sodium citrate in Pichia pastoris(Manuscript ID: jof-2250804) are highly appreciated. We have gone through the manuscript again and revised it carefully according to your suggestions. All the changes have been highlighted in red in the text. Thus the revised version is submitted online for you to reconsider for publication in Journal of Fungi. In the following pages are our point-by-point responses to each of the comments.

We look forward to hearing from you at your earliest convenience.

Yours sincerely,

Lin Liangcai

______________________________

Liangcai Lin (PhD, Associate professor)

Tianjin University of Science and Technology

College of Bioengineering, Tianjin 300457, China

E-mail: lclin@tust.edu.cn.

Responses to the comments of Reviewer #2:

  • 1-The abstract is not highlighting the main results that you revealed in the manuscript;

Response: Thanks for your suggestion. We have rewritten the abstract and highlighted the main results of this study, please see lines 25-27 and lines 29-32 in the revised manuscript.

  • 2- In figure 1a, you presented a 3D structure, but you don't mention anything about it, is a simulation is from RCSB, change was made at the aminoacid level, did this change /point mutation affects the 3D structure?

Response: In this study, the corresponding base sequence was input in the computer simulation, and no significant changes were observed in the 3D structure of AOX1 enzyme vary from R464 to K464. Therefore, only the change of amino acid in AOX enzyme structure were mentioned in this paper.

  • 3- You used a Delft medium for several cultivation experiments but I couldn't find any reference or detailed composition;

Response: Delft basic salt medium used for cell cultivation was consisted of 14.4 g/L KH2PO4, 7.5 g/L (NH4)2SO4, 0.5 g/L MgSO4•7H2O, 1 ml/L vitamin solution, 40 mg/L histidine, and 2 ml/L trace metal solution. We have provided this information and related references in the revised manuscript, please see lines 114-116.

References:

Zhou, Y.J., Buijs, N.A., Zhu, Z., Gómez, D.O., Boonsombuti, A., Siewers, V., and Nielsen, J. 2016. 'Harnessing yeast peroxisomes for biosynthesis of fatty-acid-derived biofuels and chemicals with relieved side-pathway competition', Journal of The American Chemical Society, 138: 15368-15377.

Zhou, Y.J., Buijs, N.A., Zhu, Z., Qin, J., Siewers, V., and Nielsen, J. 2016. 'Production of fatty acid-derived oleochemicals and biofuels by synthetic yeast cell factories', Nature Communications, 7: 11709.